# Injectable Networks Based on a Hybrid Synthetic/Natural Polymer Gel and Self-Assembling Peptides Functioning as Reinforcing Fillers

**DOI:** 10.3390/polym15030636

**Published:** 2023-01-26

**Authors:** Alina Ghilan, Alexandra Croitoriu, Aurica P. Chiriac, Loredana Elena Nita, Maria Bercea, Alina Gabriela Rusu

**Affiliations:** Petru Poni Institute of Macromolecular Chemistry, 41 A Grigore Ghica Voda Alley, 700487 Iasi, Romania

**Keywords:** double network hydrogels, low-molecular-weight gelators, sodium alginate, poly(itaconic anhydride-co-3,9-divinyl-2,4,8,10-tetraoxaspiro (5,5) undecane)

## Abstract

Double network (DN) hydrogels composed of self-assembling low-molecular-weight gelators and a hybrid polymer network are of particular interest for many emerging biomedical applications, such as tissue regeneration and drug delivery. The major benefits of these structures are their distinct mechanical properties as well as their ability to mimic the hierarchical features of the extracellular matrix. Herein, we describe a hybrid synthetic/natural polymer gel that acts as the initial network based on sodium alginate and a copolymer, namely poly(itaconic anhydride-co-3,9-divinyl-2,4,8,10-tetraoxaspiro (5,5) undecane). The addition of amino acids and peptide-derived hydrogelators, such as Fmoc-Lys-Fmoc-OH and Fmoc-Gly-Gly-Gly-OH, to the already-made network gives rise to DNs crosslinked via non-covalent interactions. Fourier transform infrared spectroscopy (FTIR) and thermal analysis confirmed the formation of the DN and highlighted the interactions between the two component networks. Swelling studies revealed that the materials have an excellent water absorption capacity and can be classified as superabsorbent gels. The rheological properties were systematically investigated in response to different variables and showed that the prepared materials present injectability and a self-healing ability. SEM analysis revealed a morphology consisting of a highly porous and interconnected fibrous network. Finally, the biocompatibility was evaluated using the MTT assay on dermal fibroblasts, and the results indicated that the new structures are non-toxic and potentially useful for biomedical applications.

## 1. Introduction

Short peptides and amino acids have received increasing interest as molecular blocks in various medicinal fields due to their superior biocompatibility, tunable biodegradability, low toxicity, and cost. Among the most attractive applications of peptides, self-assembling supramolecular hydrogels are particularly fascinating for drug delivery, regenerative medicine, or 3D cell culture systems [1]. Supramolecular peptide-based hydrogels consist of physical entanglements of filamentous assemblies caused by various types of non-covalent interactions, such as hydrogen bonds, hydrophobic interactions, electrostatic interactions, and π-π interactions [2]. 

They have a unique reversibility that is mostly lacking in chemically crosslinked hydrogels [3]. Furthermore, various bioactive principles can be incorporated into the network, which can also provide a stimulus-triggered and well-controlled release capability [4]. L-lysine (Lys) is a physiologically occurring, bifunctional amino acid known for its potential to support injury repair, playing an important role in cell adhesion and collagen crosslinking [5]. Another essential amino acid is glycine (Gly), which has an antioxidant role, with studies demonstrating a significant reduction in oxidative stress markers in the infarcted regions of the brain [6]. In order to control the reactions of the peptides, they are modified at the N–terminal with a voluminous aromatic group, the most common of which is 9-fluorenylmethyloxycarbonyl (Fmoc) [7]. Fmoc acts as a protecting group while its aromatic rings promote hydrophobic and π-π stacking interactions [8]. Studies have shown that these two amino acids (either modified or unmodified) have the ability to act as low-molecular-weight gelators (LMWG), serving as an excellent platform for the preparation of gels due to their ability to self-assemble [7,9]. Thus, hydrogels based on Lys or Gly have been successfully obtained, with applications in cell cultures, as vehicles for the sustained delivery of drugs, for efficient dye removal, as materials with antibacterial activity against Gram-positive and Gram-negative bacteria, or in musculoskeletal tissue engineering [10,11,12,13]. However, the majority of these gels are typically mechanically weak and brittle, which restricts their range of uses for applications requiring higher mechanical properties [14].

We have recently reported the synthesis of a new copolymer, namely poly(itaconic anhydride-co-3,9-divinyl-2,4,8,10-tetraoxaspiro (5,5) undecane) (PITAU). PITAU has the ability to form networks through specific functional groups, it is biodegradable and biocompatible, it has binding properties, amphiphilicity, and thermal stability, but also sensitivity to pH and temperature [15]. This polymer is especially well suited for biomedical applications due to its characteristics, versatility, and untapped potential [16]. In the context of our previous research, the possibility of forming biocompatible hydrogels with superior properties based on PITAU and a natural polymer, alginate (Alg), was investigated. Thus, we obtained a chemically crosslinked hydrogel (PITAU/Alg) by opening the anhydride cycle of the copolymer with the hydroxyl groups of the polysaccharide [17]. The prepared compounds were further characterized, highlighting the formation of a 3D network together with the ability of the new structure to release therapeutic substances in a controlled manner. 

In the present work, we extend our previous studies to improve the PITAU/Alg gel system by adding an amino acid/peptide capable of gelation and formation of supramolecular structures, by which an increased functionality is ensured for the newly formed double network (DN). DN hydrogels can exhibit exceptional properties due to their unique contrasting network structures and strong interpenetrating entanglements [18]. They can also recreate the molecular and spatiotemporal complexity of the native extracellular matrix within a single network and allow the controlled release of small molecules [19,20]. Although DNs have already been obtained from a variety of synthetic and natural polymers, the use of peptides in obtaining such multicomponent systems is quite recent [21,22]. 

To establish physical interactions between the peptide network and the preformed hydrogel matrix (PITAU/Alg), Fmoc-Lys-Fmoc-OH amino acid and Fmoc-Gly-Gly-Gly-OH tripeptide were chosen as co-partner systems. In addition, we systematically examined how peptides influence the rheological, thermal, and morphological characteristics of the DN hydrogels The obtained results demonstrated that these multicomponent materials show injectability and the ability to self-heal and recover their structural integrity. Considering the high demand for the development of biocompatible systems that can bypass the need for systemic drug administration, the proposed materials can find applicability in the controlled delivery of therapeutics as well as in tissue engineering. 

## 2. Materials and Methods

### 2.1. Materials

All chemicals used were reagent-grade and used as purchased without further purification. Fmoc–Lys–Fmoc–OH amino acid (C_36_H_34_N_2_O_6_, *M*_w_ = 590.66 g/mol), 3,9-divinyl-2,4,8,10-tetraoxaspiro[5.5] undecane (U) (purity 98%), itaconic anhydride (ITA) (purity 98%), and 2,20-Azobis (2-methylpropionitrile) (AIBN) (purity 98%) were purchased from Sigma-Aldrich (Darmstadt, Germany). Fmoc-Gly-Gly-Gly-OH tripeptide (C_21_H_21_N_3_O_6_, Mw = 411.41 g/mol) was obtained from Bachem (Bubendorf, Switzerland). Alginic acid sodium salt from brown algae was supplied by Acros Organics (Geel, Belgium). Sodium phosphate buffer (PBS, pH 7.4, 0.01 M) was prepared using monosodium phosphate (NaH_2_PO_4_ × 2H_2_O) and disodium phosphate (Na_2_H_2_PO_4_ × 7H_2_O) via standard protocol. Dimethyl sulfoxide (DMSO) was acquired from Fluka (Buchs, Switzerland). Dulbecco’s Modified Eagles Medium (DMEM) high glucose (4500 mg/L glucose with L-glutamate and pyruvate), fetal bovine serum (FBS), antibiotics mixture (penicilin-streptomycin-neomycin mixture-P/S/N ≈5000 units penicilin, 5mg streptomycin, 10 mg neomycin/mL), and thiazolyl blue tetrazolium bromide (MTT) were purchased from Sigma-Aldrich (Steinheim, Germany), while the nutrient mixture HAM-F12 was purchased from Biological Industries, Beit-Haemek, Israel. The water used in the experiments was purified using an Ultra Clear TWF UV System.

### 2.2. Preparation of the DN Hydrogels

The DN gel design used for this study consists of two parts: (i) the preparation of a chemical pre-gel based on PITAU synthetic copolymer and sodium alginate (PITAU/Alg), which constitutes the system's first network, and (ii) the addition to the PITAU/Alg of peptides with the ability to self-assemble into nanofibrous morphologies, which generates the system's second network (Figure 1). 

As mentioned, the first stage consists of the formation of the hybrid synthetic/natural polymer gel that acts as the initial network. For this, the PITAU copolymer was used, of which a detailed description of the preparation method is already available [15]. In brief, the copolymer was synthesized through a continuous radical process of polymerization with a ratio of ITA/U = 1.5/1 between comonomers using AIBN as an initiator (0.9%) and 1,4-dioxane as a solvent. The pre-gel was obtained by mixing the PITAU solution in dioxane with an aqueous alginate solution (3 wt.%) at a gravimetric ratio of 1/2 [17].

For the second step, two solutions based on Fmoc-Lys-Fmoc-OH 1 wt.% and Fmoc-Gly-Gly-Gly-OH 1 wt.% were prepared in DMSO using the aprotic polar solvent-triggered gelation technique. An amount of 0.01 M phosphate buffer (pH = 7.4) was added gradually. Subsequently, the two individually prepared solutions were mixed in a gravimetric ratio of 1:1 and 1:3 resulting in co-assemblies coded P1 (Fmoc-Lys-Fmoc-OH:Fmoc-Gly-Gly-Gly-OH 1:1) and P2 (Fmoc-Lys-Fmoc-OH:Fmoc-Gly-Gly-Gly-OH 1:3). These co-assemblies in different ratios were used in the present study because the combination of two different types of peptides in a single architecture can lead to a synergy of their properties or new functionalities, which are worth investigating. 

Therefore, four DN hydrogels were obtained by adding predetermined amounts of the Fmoc-Lys-Fmoc-OH, Fmoc-Gly-Gly-Gly-OH, and P1 and P2 co-assemblies over the PITAU/Alg pre-gel as presented in Table 1. The compounds were slowly mixed for 5 min and then the obtained DNs were left to mature for 24 h. Finally, the samples were lyophilized, purified by dialysis with deionized distilled water until the conductivity was close to 0, lyophilized again, and characterized.

### 2.3. Samples Characterization

#### 2.3.1. Fourier Transform Infrared Spectroscopy (FTIR) Analysis

FTIR spectra of the studied materials were recorded on a Vertex Bruker Spectrometer (Bremen, Germany) in an absorption mode ranging from 400 to 4000 cm^−1^. The samples were ground with potassium bromide (KBr) powder and compressed into a disc for analysis. The spectra were generated at 4 cm^−1^ resolution with an average of 64 scans. 

#### 2.3.2. Rheological Measurements

The rheological measurements were carried out at 25 °C by using a MCR 302 Anton-Paar rheometer (Graz, Austria) equipped with plane–plane geometry (the diameter of the upper plate of 50 mm and the gap of 0.5 mm) and a Peltier system for temperature control.

The linear range of viscoelasticity was established for each sample in amplitude sweep tests and in all cases it was reached from very low strain values (γ = 0.01%) up to a minimum γ of 5%. The oscillatory shear measurements were performed as a function of the oscillation frequency (ω), from 0.1 rad/s to 100 rad/s, for a constant strain of 1%. The elastic (G′) and viscous (G″) moduli were determined, indicating the stored and dissipated energy during one cycle of deformation, respectively. The loss tangent (tanδ = G″/G′) is correlated with the degree of viscoelasticity of each sample (tanδ > 1 is characteristic to viscous flow; tanδ < 1 suggests solid-like behavior).

The structure recovery (self-healing ability) was studied in oscillatory shear experiments. A constant ω value of 10 rad/s was set and the strain amplitude was systematically changed during successive runs from low (1%), high (50%, 100%, 200%, 500%, 1000%), and low again (1%) values, respectively. G′, G″, and tanδ were measured as a function of time.

The apparent viscosity (η) of the samples was determined in stationary shear flow conditions, for shear rates ($\dot{\gamma }$) varying in the range 0.01–100 s^−1^.

#### 2.3.3. Swelling Study

The swelling degree of the DN hydrogels was determined by immersing the samples in phosphate-buffered solution (PBS 0.01 M, pH = 7.4) at room temperature (25 °C). Freeze-dried samples were weighed (*W*_d_) and placed in the swelling environment. The amount of absorbed solution was monitored gravimetrically. The swollen hydrogels were taken out from the swelling medium at regular intervals, dried superficially with filter paper, and weighed in the swollen state (*W*_s_) until a constant weight was reached. The swelling experiments were performed in triplicate for each sample. The equilibrium swelling degree (SDE) was determined as follows:SDE(%) = (*W*_s_ − *W*_d_)/*W*_d_ × 100 (1)
where *W*_s_ is the weight of the swollen samples at equilibrium and *W*_d_ is the weight of the freeze-dried samples.

#### 2.3.4. Thermal Analysis

The thermal behavior of the samples was studied using an STA 449 F1 Jupiter thermo-balance made by Bruker, Bremen, Germany. The dried compounds weighing from 10 to 12 mg were placed in an open Al_2_O_3_ crucible and heated in dynamic mode from room temperature up to 675 °C. Runs were performed with a heating rate of 10 °C/min in a nitrogen atmosphere with a gas flow of 40 mL/min. Differential thermal analysis was recorded simultaneously on the same apparatus using Al_2_O_3_ as the reference material. Data collection and processing were performed using Proteus 5.0.1 software.

#### 2.3.5. Scanning Electron Microscopy (SEM) Analysis

The microstructural architecture of the freeze-dried DN hydrogels was analyzed by using SEM microscopy. The investigations were performed on samples fixed in advance by means of colloidal copper supports. The samples were spray-coated with a thin layer of gold. The covered area was examined by using a scanning electron microscope, type Quanta 200, which operates at 30 kV with secondary electrons in high vacuum mode.

#### 2.3.6. In Vitro Biocompatibility Assay MTT

In vitro cytotoxicity of the DN hydrogels was determined by using the MTT assay. Briefly, the hydrogels were sterilized by UV light for 30 min and then immersed in complete DMEM-HAM F12 medium (10% BFS, 1% PSN). The medium was replaced with fresh medium until its color remained unchanged. Next, the hydrogels were incubated for 24 h at 37 °C for equilibration. After that, 100 µL of rabbit dermal fibroblast cell suspension (1 × 10^4^ cells/100 µL) was added to each well of a 24-well plate. The culture plates were incubated in 5% CO_2_ atmosphere at 37 °C for 24 h. After the cell incubation period, the prepared hydrogels (DMEM-HAM F12 medium + hydrogel) were added to the cells, then incubated in 5% CO_2_ atmosphere at 37 °C for 96 h. Wells with cells without the hydrogel were chosen as controls. The cytotoxicity was assessed using 3-[4,5-dimethylthiazole-2-yl]-2,5-diphenyltetrazolium bromide (MTT) assay. Finally, 5% MTT was added to each well for the last 3 h at 37 °C and then a solubilization solution (DMSO) was added to dissolve the formazan crystals into a purple-colored solution. The absorbance (*Abs*) was measured at 570 nm using a Tescan Sunrise Plate Reader. Cell viability was determined by using the formula
(2)$$\mathrm{viability}\left(\%\right)=\frac{Abs_{sample}}{Abs_{control}}\times 100$$

The data from the cytotoxicity assay was reported as a percentage of the control absorbance. MTT assay was performed in triplicate.

## 3. Results

### 3.1. Fourier Transform Infrared Spectroscopy (FTIR) Analysis

The PITAU/Alg pre-gel spectrum combines both the characteristic bands of the copolymer and those of the polysaccharide (Figure 2a). A detailed analysis of the FTIR spectrum of this hybrid gel is already described in previous studies [17]. 

In brief, the synthetic copolymer's characteristic bands are at 1780 cm^−1^ and 1856 cm^−1^ for the C=O symmetric and asymmetric stretching of the anhydride unit, 1660 cm^−1^ corresponding to C=C stretching, 1400 cm^−1^ attributed to =CH_2_ in plane deformation. A broad signal at 1080 cm^−1^, corresponding to the C–O–C ether stretch that confirms the presence of spiroacetal fragments, is also maintained in the PITAU/Alg spectrum. Furthermore, the spectrum of the gel closely follows the presence of the main vibrations of alginate. The most important absorption bands of the natural polymer are found at 3494 cm^−1^, which is specific to the stretching vibrations of the OH group, 2938 and 2853 cm^−1^ for the asymmetric and symmetric stretching vibrations of the C–H bond, respectively, and 1626 cm^−1^ attributed to the stretching of the C=O bond, also confirmed by other studies [23]. The synthesis of the hydrogel is confirmed by the disappearance of the bands at 1856 cm^−1^ and 1780 cm^−1^, which indicates the formation of chemical bonds through the opening of the itaconic anhydride ring with the OH groups of the alginate.

Comparing the synthesized DN gels spectra (Figure 2b) with those of the precursors, it is observed that the chemical structure of the PITAU/Alg is maintained in all samples. Additionally, the structural similarity of the constituent components leads to overlaps of the characteristic bands in the DN spectrum. Therefore, the specific bands of the respective Lys amide group vibrations and O–H bond vibrations overlap with the specific bands of the O–H or C=O vibrations from the pre-gel. Consequently, the differences between the spectra of the precursors and those of the prepared samples are reduced to minor shifts in the various bands and variable levels of the intensity of the absorption. Furthermore, the presence of a Gly peptide, which contains functional groups similar to those of the amino acid, emphasizes the previously described differences. The most significant differences are recorded in the absorption region of the O–H bond, which is shifted to higher values, attributed to the formation of hydrogen bonds both between the pre-gel/peptides and also between the two peptide co-partners Lys and Gly. These shifts in the absorption bands confirm the formation of the DN hydrogels.

### 3.2. Rheological Behavior

Figure 3 shows an example of the variation in viscoelastic parameters for the sample PITAU/Alg*P2 as a function of strain amplitude or oscillation frequency.

The plots show a network structure, G′ > G″ and tanδ ≈ 0.34. Small variations of G′ and G″ observed in Figure 3b are attributed to dynamics of intermolecular interactions under oscillatory deformations. The hydrogel strength increases as the content of Gly increases, thus G′ for PITAU/Alg*P2 exceeds the value of G′ registered for PITAU/Alg (Figure 4). This fact was expected considering the more numerous possibilities of forming physical bonds of Gly through its available functional groups, compared to Lys which can make fewer intermolecular bonds due to the steric hindrances generated by the presence of the two bulky Fmoc groups. The lowest values of G′ are obtained for PITAU/Alg samples in the presence of one peptide, Gly interactions with polymer chains being slightly higher compared with Lys: these two samples present G″ > G′ (liquid-like behavior). Another observation concerns the slope of G′ and ω dependences, from approx. 0.7–0.8 for PITAU/Alg in the presence of one peptide to 0.15 for PITAU/Alg or PITAU/Alg*P2. In the case of G″, the slopes vary between 0.67 (PITAU/Alg*Lys or PITAU/Alg*Gly) and 0.14 (PITAU/Alg*P2).

Shear-thinning behavior is an important feature for applications of injectable hydrogels, i.e., the viscosity decreases when a shear flow is applied to materials. The prepared hydrogels present injectability, the viscosity decreases as the shear rate increases above 0.04 s^−1^ (Figure 5). In the non-Newtonian region, the apparent viscosity (η) scales as ${\dot{\gamma }}^{-n}$, where the power law index, *n*, varies between 0.2 (sample PITAU/Alg*Lys) and 0.6 (PITAU/Alg*P2 or PITAU/Alg samples). More than that, the DN hydrogels injected into the body are able to quickly self-heal after the external force is removed (Figure 6).

The thixotropy was carefully investigated following the time dependence of the viscoelastic parameters (G′, G″, and tan δ) when cycles of successive low and high levels of deformations were alternatively applied every 300 s to the hydrogels: 1% (this value belongs to linear domain of viscoelasticity), 50%/100%/200%/500%/1000% (these strain values are in the nonlinear domain of viscoelasticity), and again 1% (as an example, Figure 7 shows the behavior of the sample PITAU/Alg*P2). The recovery of the network structure, as monitored by G′ and G″ values during the test, is very fast when the deformation is suddenly changed from high to low values. G″ values remain almost unchanged after five cycles of increasing deformations, where G′ slightly increases after each cycle of deformation as a consequence of molecular rearrangements during high deformations.

The samples PITAU/Alg*P1 and PITAU/Alg present similar behavior as shown in Figure 7, whereas samples PITAU/Alg*Lys and PITAU/Alg*Gly recover their structure only after one cycle from 1% to 50% and again to 1%; for higher deformations, the samples were not able to recover their structure.

### 3.3. Swelling Study

Hydrogels placed in an aqueous environment allow water molecules to diffuse into the network, and this affinity for water varies depending on the DN composition and architecture. As can be seen in Figure 8, all the tested materials have a superabsorbent character [24].

The chemical nature of the used compounds has a significant impact on the formation of highly hydrophilic materials and in determining the superabsorbent properties. Therefore, upon exposure of PITAU/Alg pre-gel to PBS at pH = 7.4, the carboxyl groups (COOH) of the polysaccharide are ionized into carboxylate groups (COO−), and as a result, anion-anion electrostatic repulsion forces are formed in the hydrogel that increase the distance between the polymer chains and allow water diffusion into the matrix [25]. Next, the inclusion of the second network based on amino acids/peptides in the DN hydrogel highly increases the equilibrium swelling degree (SDE). This can be explained by SEM microscopy which reveals for the DN systems a very porous three-dimensional network interspersed with a fibrous network that allows a better diffusion of water molecules compared to the PITAU/Alg pre-gel which has a denser and inhomogeneous morphology with small or collapsed pores. Interestingly, the SDE order (from high to low) for the four DNs was PITAU/Alg*Lys > PITAU/Alg*P1 > PITAU/Alg*P2 > PITAU/Alg*Gly, indicating that a higher Lys content allows for more water to be absorbed. As shown by the rheology data, the addition of Gly to the DN systems resulted in hydrogels with a tighter network due to the formation of more physical interactions through its available functional groups which, in turn, leads to less swelling due to the hindered mobility of the chains. The presence of Lys leads to the formation of fewer intermolecular bonds due to the steric hindrance generated by the presence of the two aromatic moieties and, consequently, allows the movement of water molecules in the network.

Finally, these findings show that the hydrophilicity of DN hydrogels can be controlled by composition. In addition, the superabsorbent character may be beneficial for drug incorporation by diffusion as well as in tissue engineering applications where such materials may allow cells to migrate and proliferate during regeneration by mimicking their natural aqueous environment.

### 3.4. Thermal Analysis

The thermal behavior of the DN hydrogels based on a hybrid pre-gel interpenetrated with amino acids/peptides was investigated and it is presented in Figure 9, while in Table 2 the main thermal parameters are presented. PITAU/Alg has five stages of decomposition, whereas the DN systems have four stages of decomposition. In the case of PITAU/Alg, the mass loss recorded in the first stage (~4.70%) is associated with the evaporation of water from the network. The second decomposition process shows the highest mass loss (28.48%) due mainly to the breakage of the OH, COOH, and COO- functional groups, with a T_peak_ at a temperature of 184°C. The third stage of decomposition with a T_peak_ at 230 °C and a mass loss of 15% occurs as a result of the decomposition of the glycosidic bonds in the polysaccharide structure [26]. According to the study by Shen et al., the decomposition of spiroacetal-type structures occurs in the range of 210–349 °C [27]. Therefore, stage IV with a T_peak_ at 265 °C and a mass loss of ~11% corresponds to the degradation of the spiroacetal fragments in the PITAU copolymer structure. The last degradation process with a T_peak_ at 325 °C occurs as a result of the decarbonylation reaction of sodium alginate, forming aliphatic derivatives and releasing CO_2_.

By adding Lys, Gly, or different ratios of the amino acid/peptide co-assembly solutions over the first gel network, thermal stability is increased due to the formation of new physical bonds between complementary functional groups. As a result, the DTG curves show that the DN systems undergo thermal decomposition in four stages. According to the literature [28], amino acids thermally decompose to form H_2_O, CO_2_, CO, NO_x_, and NH_3_, and lactams or heterocyclic compounds with -CO-NH bonds are found in their residue [29]. Table 2 shows that the first decomposition process has a T_peak_ below 150°C and a rapid mass loss (4.33–5.11%) and occurs as a result of the removal of H_2_O and residual solvents. For the second process, T_peak_ was recorded at ~170 °C, which is characteristic of the cleavage and decomposition of side groups in the structure of Lys and Gly tripeptides. Increasing the temperature above 250 °C generates a major mass loss determined by the decomposition of the carbamate group from the Fmoc structure, the decomposition of the spiroacetal structure from PITAU, and the decomposition of the pyranose ring from alginate. The last decomposition process showing a T_peak_ at ~390 °C is attributed to the decomposition of the fluorenyl aromatic rings in the Fmoc structure. The amount of residue (27.51%) and the T_50_ parameter (315 °C) indicate that the PITAU/Alg*Lys-interpenetrated hydrogel is the most thermally stable, owing to the presence of the Fmoc group at both ends of the amino acid structure. Furthermore, the DN system containing amino acids/peptides co-assembled in the ratio 1:3 shows a higher amount of residue ~20.81% and T_50_ at 286 °C compared to the system containing amino acids/peptides co-assembled in a 1:1 ratio (15% residue and T_50_ at 275 °C). This difference can be attributed to the formation of intramolecular and intermolecular H-bonds due to the presence of more -CO-NH- bonds in the Gly tripeptide.

### 3.5. Microscopy Analysis

SEM was used to assess the morphology of the DN hydrogels but also to understand the relative differences in microstructure after the addition of the peptides and formation of DN gels. The results are displayed in Figure 10. Thus, PITAU/Alg gel exhibits a dense and inhomogeneous structure with very small or collapsed pores. All four DN hydrogels show a different morphology given by the predominantly fibrous nature of the amino acid/peptide. The synthesized compounds present a structure with large pores interspersed with a fibrous network. The sample with Gly (PITAU/Alg*Gly) and Lys (PITAU/Alg*Lys) show a morphology consisting of highly interconnected fibrous networks, whereas the two DNs that have as the second network the amino acid/peptide co-assemblies (PITAU/Alg*P1 and PITAU/Alg*P2) in different ratios, were composed of more homogeneous porous structures with interlaced fibers. These results indicate that the use of the co-assembly could change the molecular arrangement during the assembly process. Based on the SEM images, we can conclude that the second amino acid/peptide self-assembled network was successfully integrated into the first network constituted by PITAU/Alg by inducing strong hydrogen bonds, thus leading to the successful obtaining of a DN system. 

### 3.6. In Vitro Biocompatibility Assay 

To identify whether the studied materials were cytotoxic, skin fibroblasts were exposed to the DN hydrogels, while cell viability was quantified using the standard MTT assay. Experiments revealed that the DN hydrogels gave > 75% cell viability after 96 h culture (Figure 11). These results suggest that the prepared compounds could be considered biocompatible because, according to the specialized literature, a material is considered cytocompatible if the percentage of viable cells is equal to or greater than 70% [30,31,32]. To conclude, in terms of material safety, the proposed DN systems can thus be considered potentially non-toxic, holding promise as injectable materials for biomedical applications, such as tissue engineering.

## 4. Conclusions

In summary, this research focused on the preparation of new DN hydrogels as well as their comprehensive characterization for their further use in biomedical applications. In this sense, a gel based on PITAU and alginate, as a hybrid network that can maintain the integrity of the material over time, was combined with a secondary network formed by low molecular weight gelators, such as Fmoc-Lys-Fmoc-OH and Fmoc-Gly-Gly-Gly-OH to improve the functionality and biocompatibility. The formation of hydrogen bonds between the pre-gel structure and peptides was highlighted in the FTIR spectra by shifts to higher values of the absorption bands characteristic of hydroxyl groups. These results are also supported by the thermal analyses that reveal an increased stability in the case of DN as a result of the formation of new physical bonds between complementary functional groups. Swelling studies revealed that the materials have an excellent water absorption capacity and can be classified as superabsorbent gels. 

Furthermore, one of the advantages of this strategy is that the properties can be controlled by changing the type of amino acid/peptide in the system, which can lead to increasing the number of physical interactions in these DNs. Therefore, the design proposed in this study provides DN hydrogels with better rheological properties than can be obtained from a single-component network. The strongest DN gel proved to be composed of PITAU/Alg*P2, Fmoc-Gly-Gly-Gly-OH interactions with the polymer chains being slightly higher compared with Fmoc-Lys-Fmoc-OH. Moreover, the data show that all the compounds studied present injectability and exhibit thixotropic behavior, with the results showing that the recovery of the network structure is very fast when the strains are suddenly changed from high values to low values. Based on the SEM images, we can conclude that the obtained gels present a structure with large pores interspersed with a fibrous network. Finally, DN hydrogels were found to be cytocompatible, indicating that the proposed compounds could be suitable options for the controlled delivery of therapeutics or tissue engineering.

## Figures and Tables

**Figure 1 polymers-15-00636-f001:**
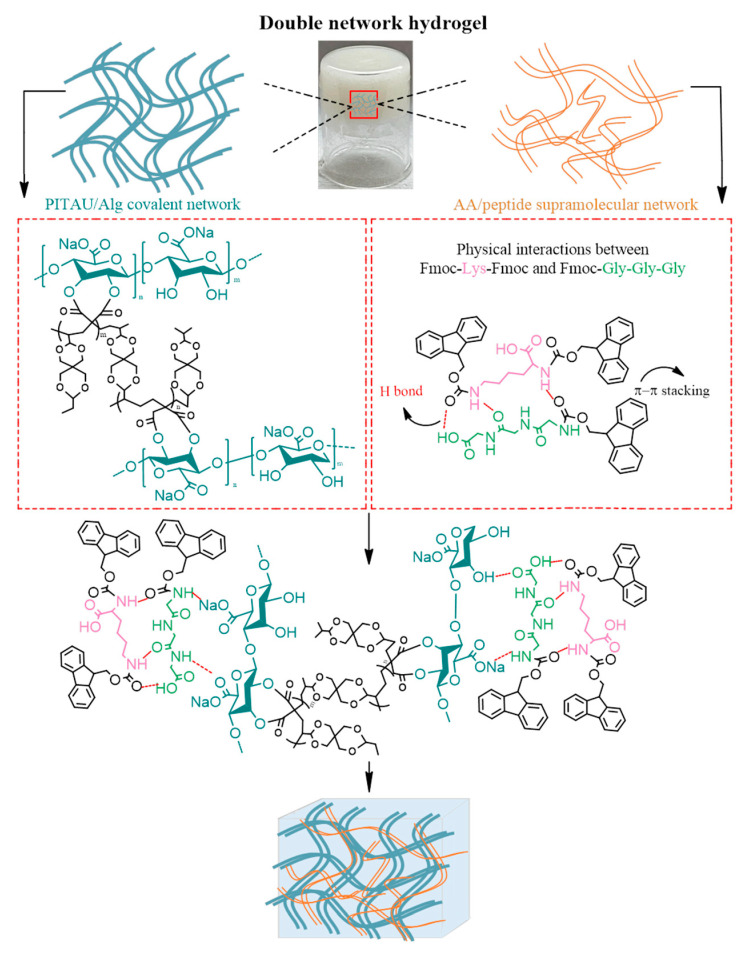
Illustrative scheme of the DN hydrogel formation with PITAU/Alg covalent gel as the first network and self-assembled peptides as the second network.

**Figure 2 polymers-15-00636-f002:**
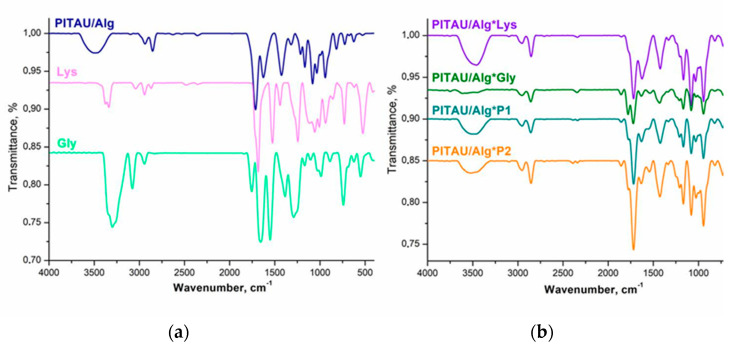
FTIR spectra of (**a**) PITAU/Alg, Lys, Gly and (**b**) the synthesized DN hydrogels.

**Figure 3 polymers-15-00636-f003:**
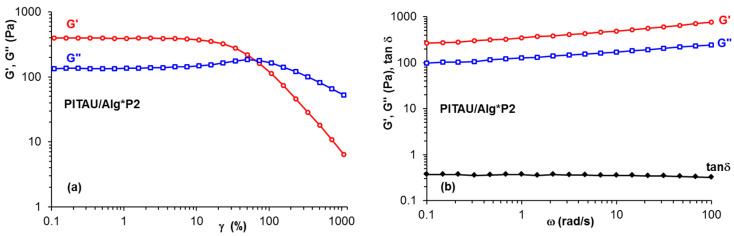
The dependence of the viscoelastic parameters for the sample PITAU/Alg*P2 in (**a**) amplitude sweep tests and (**b**) frequency sweep tests.

**Figure 4 polymers-15-00636-f004:**
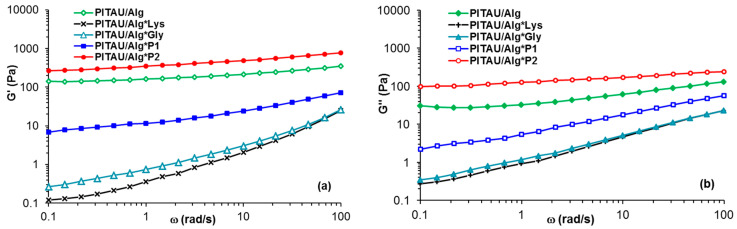
The elastic (**a**) and viscous (**b**) moduli as a function of the oscillation frequency for the investigated samples (γ = 1%).

**Figure 5 polymers-15-00636-f005:**
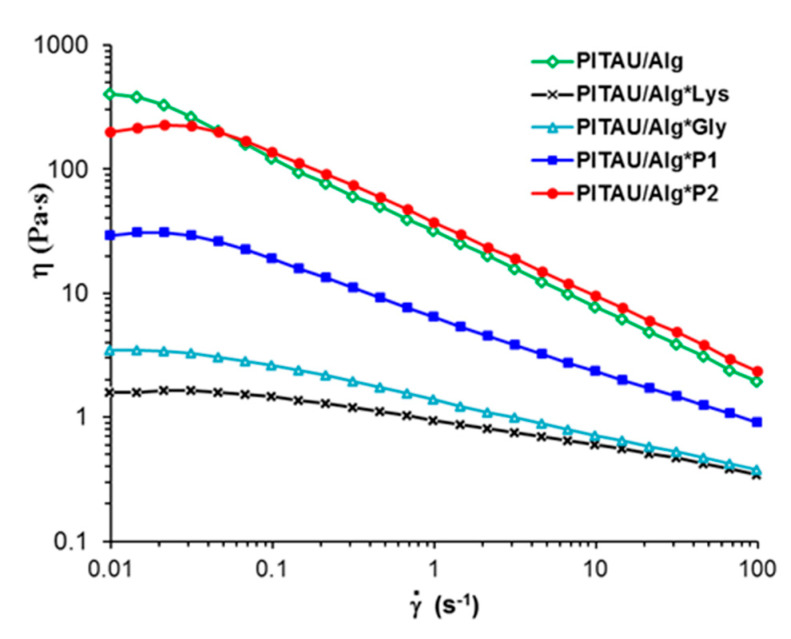
The apparent viscosity as a function of the shear rate for the samples investigated in stationary shear flow conditions.

**Figure 6 polymers-15-00636-f006:**
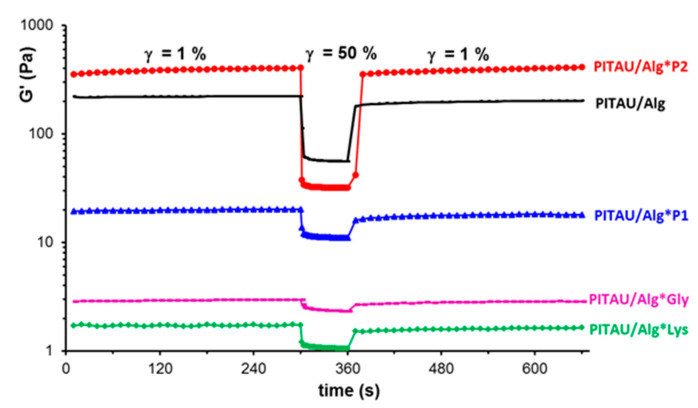
Illustration of the self-healing behavior through the dependence of G′ as a function of time during consecutive step strain measurements at low (1%), high (50%), and low (1%) values of deformation (γ).

**Figure 7 polymers-15-00636-f007:**
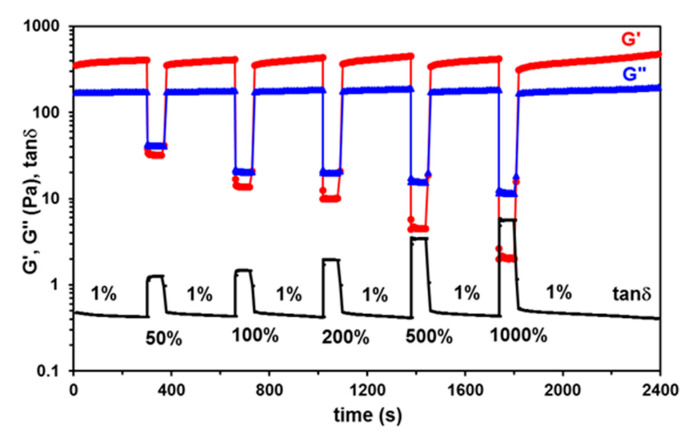
The self-healing ability of the sample PITAU/Alg*P2. The hydrogel was submitted to consecutive cycles of step strains at low (1%) and high (50%; 100%; 200%; 500%; 1000%) and low (1%) strains.

**Figure 8 polymers-15-00636-f008:**
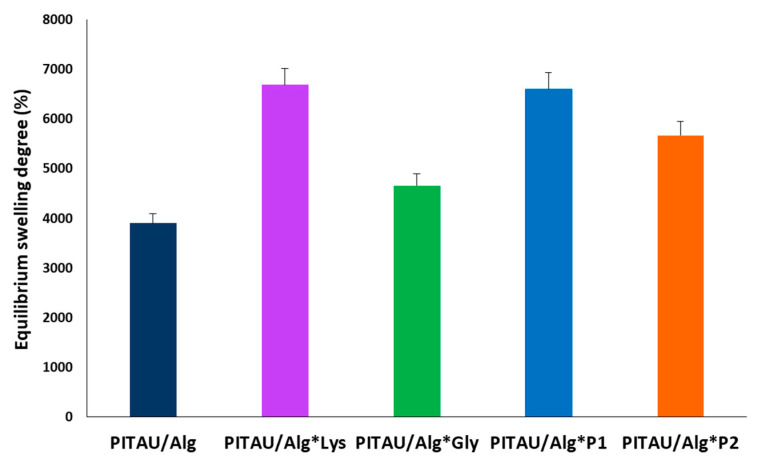
Equilibrium swelling degree of PITAU/Alg and DN hydrogels.

**Figure 9 polymers-15-00636-f009:**
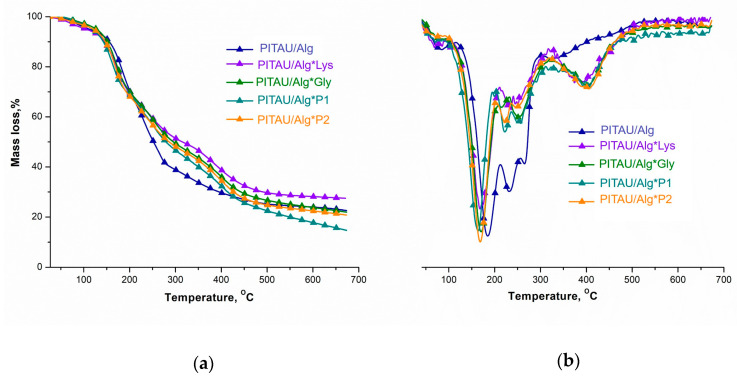
(**a**) TG and (**b**) DTG curves of the DN hydrogel and PITAU/Alg pre-gel.

**Figure 10 polymers-15-00636-f010:**
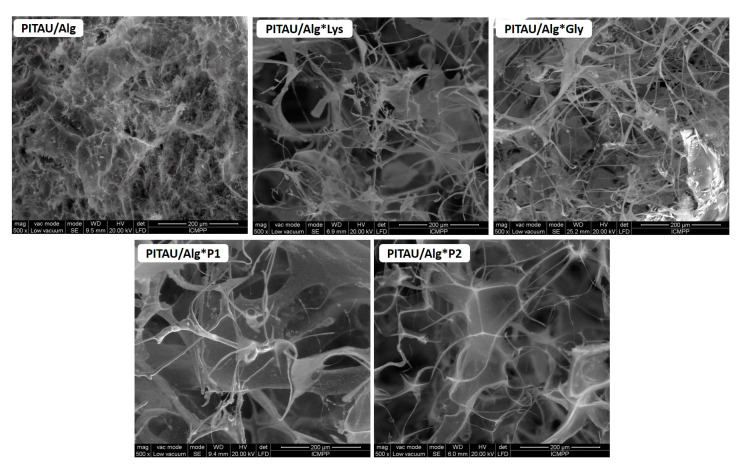
SEM microscopy of the PITAU/Alg and DN hydrogels.

**Figure 11 polymers-15-00636-f011:**
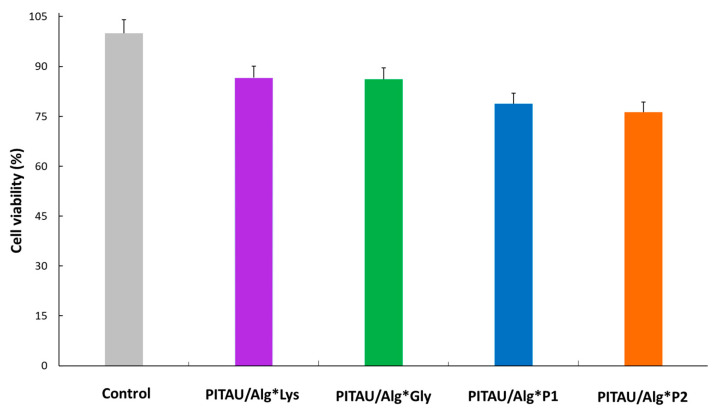
Viability of human fibroblast cells evaluated by MTT assay at 96 h. All data are expressed as a percentage with respect to the untreated control cells and are the mean of three independent experiments ± standard error of the mean (SEM).

**Table 1 polymers-15-00636-t001:** The composition of the samples evaluated in the study.

Sample	Samples Composition					
	Alginate (g)	PITAU (g)	Lys (g)	Gly (g)	DMSO (mL)	PBS (mL)
**PITAU/Alg**	6.72	3.6	-	-	-	-
**PITAU/Alg*Lys**	6.72	3.6	0.1	-	0.48	9.52
**PITAU/Alg*Gly**	6.72	3.6	-	0.1	0.48	9.52
**PITAU/Alg*P1**	6.72	3.6	0.05	0.05	0.48	9.52
**PITAU/Alg*P2**	6.72	3.6	0.025	0.075	0.48	9.52

**Table 2 polymers-15-00636-t002:** The main thermal parameters obtained by thermal degradation of gels.

Sample	Degradation Stage	T_onset_ (°C)	T_peak_ (°C)	T_endset_ (°C)	W (%)	T_10_ (°C)	T_50_ (°C)
**PITAU/Alg**	I II III IV V	71 148 220 260 303	84 184 230 265 325	107 209 246 279 347	4.70 28.48 15.0 10.87 18.53 *22.55*	153	251
**PITAU/Alg*Lys**	I II III IV	50 120 235 374	103 171 254 375	110 206 336 444	5.11 26.23 20.36 20.79 *27.51*	145	315
**PITAU/Alg*Gly**	I II III IV	70 207 237 388	168 222 246 405	195 234 349 450	30.74 8.88 17.73 21.82 *20.81*	146	286
**PITAU/Alg*P1**	I II III IV	44 136 232 385	72 169 250 401	119 198 375 529	4.33 25.52 30.35 18.05 *21.92*	147	292
**PITAU/Alg*P2**	I II III IV	79 198 246 370	161 218 254 389	189 232 333 423	30.17 9.73 18.63 26.89 *15.05*	138	275

**T_onset_**—the temperature at which the degradation process begins; **T_peak_**—the temperature at which the rate of degradation is maximum; **T_10_**, **T_50_**—temperatures corresponding to 10% and 50% mass loss; **W**—mass losses up to 700 °C.

## Data Availability

Not applicable.

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
