# Peer review of "Injectable Networks Based on a Hybrid Synthetic/Natural Polymer Gel and Self-Assembling Peptides Functioning as Reinforcing Fillers"

_polymers, 2023, doi:10.3390/polym15030636_

Round 1

Reviewer 1 Report

This manuscript reports on the preparation of a kind of double-network hydrogels as well as their comprehensive properties. The manuscript was well organized and all the statements are verified by experimental results. Therefore, I recommend this manuscript to be pulished in Polymers at its present form. 

Here are some suggestions if it is helpful to further improve the quality of the manuscript.

1. How about the swelling properties of the PITAU/Alg gels and corresponding DNS gels?

2. The self-assembling capacity of the four low molecular weight gelators should be verified by TEM before they can be utilized to prepare DN gels.

3. Possible biomedical applications should be listed in the introduction section.

4. More relevant literature should be cited and compared to illustrate the novelty of the prepared hydrogels.

Author Response

Answers to the Reviewers to the manuscript entitled:

Injectable networks based on a hybrid synthetic/natural polymer gel and self-assembling peptides functioning as reinforcing fillers

We would like to thank the reviewers for their valuable comments that improved the quality of our paper. In this sense, corrections and additional analyses of the DN hydrogels were made and the manuscript was rewritten in accordance with the indications.

Reviewer 1: This manuscript reports on the preparation of a kind of double-network hydrogels as well as their comprehensive properties. The manuscript was well organized and all the statements are verified by experimental results. Therefore, I recommend this manuscript to be pulished in Polymers at its present form. 

Here are some suggestions if it is helpful to further improve the quality of the manuscript.

  1. How about the swelling properties of the PITAU/Alg gels and corresponding DNS gels?

Swelling tests were performed and the following section was inserted into the manuscript:

3.3. Swelling Study

Hydrogels placed in an aqueous environment allow water molecules to diffuse into the network, and this affinity for water varies depending on the DN composition and architecture. As can be seen in Figure 8, all the tested materials have a superabsorbent character [24].

Figure 8. Equilibrium swelling degree of PITAU/Alg and DN hydrogels.

The chemical nature of the used compounds has a significant impact on the formation of highly hydrophilic materials and in determining the superabsorbent properties. Therefore, upon exposure of PITAU/Alg pre-gel to PBS at a pH = 7.4, the carboxyl groups (COOH) of the polysaccharide are ionized into carboxylate groups (COO−) and as a result, anion-anion electrostatic repulsion forces are formed in the hydrogel that increases the distance between the polymer chains and allows water diffusion into the matrix [25]. Next, the inclusion of the second network based on amino acids/peptides in the DN hydrogel highly increases the equilibrium swelling degree (SDE). This can be explained by SEM microscopy which reveals for the DN systems a very porous three-dimensional network interspersed with a fibrous network that allows a better diffusion of water molecules compared to the PITAU/Alg pre-gel which has a denser and inhomogeneous morphology with small or collapsed pores. Interestingly, the SDE order (from high to low) for the four DNs was PITAU/Alg*Lys > PITAU/Alg*P1 > PITAU/Alg*P2 > PITAU/Alg*Gly, indicating that a higher Lys content allows for more water to be absorbed. As shown by the rheology data, the addition of Gly to the DN systems resulted in hydrogels with a tighter network due to the formation of more physical interactions through its available functional groups which would in turn lead to less swelling due to the hindered mobility of the chains. The presence of Lys leads to the formation of fewer intermolecular bonds due to the steric hindrance generated by the presence of the two aromatic moieties and, consequently, allows the movement of water molecules in the network.

Finally, these findings show that the hydrophilicity of DN hydrogels can be controlled by composition. In addition, the superabsorbent character may be beneficial for drug incorporation by diffusion as well as in tissue engineering applications where such materials may allow cells to migrate and proliferate during regeneration by mimicking their natural aqueous environment.

  1. The self-assembling capacity of the four low molecular weight gelators should be verified by TEM before they can be utilized to prepare DN gels.

The self-assembly capacity of the four low molecular weight gelators was previously verified by our group and the results were published in the article: New Fmoc-Amino Acids/Peptides-Based Supramolecular Gels Obtained through Co-Assembly Process: Preparation and Characterization. Croitoriu A, Nita LE, Rusu AG, Ghilan A, Bercea M, Chiriac AP. Polymers (Basel). 2022;14(16):3354. The article was cited in the introduction part. In short, the self-assembly capacity was verified and confirmed by several characterization techniques such as: circular dichroism, fluorescence, UV-VIS spectroscopy, dynamic light scattering technique, FT-IR, SEM, rheology and thermal analysis. In addition, to answer the review, STEM microscopy was performed to conclude about the supramolecular structure of the peptide-based networks. The morphology of the samples was investigated with a Scanning Electron Microscope type Verios G4 UC (Thermo Scientific, Czech Republic, BRNO) working in STEM mode at 15 kV, with a STEM 3+ detector (Bright-Field Mode). One drop of each suspension was placed on a carbon-coated copper grids with 300 mesh size and allowed to dry in a oven(c.a. 30 °C).

Fig. STEM images for Lys, Gly, and co-assemblies at 1:1 and 1:3 ratios.

  1. Possible biomedical applications should be listed in the introduction section.

A revised version of the introduction was included in the manuscript that also discusses potential biomedical uses.

  1. More relevant literature should be cited and compared to illustrate the novelty of the prepared hydrogels.

Following the Review’s indications, several new references were cited in this paper, both in the introduction section, where additions were made related to new relevant systems published in other studies, and in the characterization section, that supports the presented data.

Reviewer 2 Report

The manuscript entitled “Injectable networks based on a hybrid synthetic/natural polymer gel and self-assembling peptides functioning as reinforcing fillers” described a new copolymer PITAU and a DN hydrogel. The characterizations were well done and the results showed the DN hydrogel injectable and self-healing. MTT assay demonstrated the DN hydrogel was no toxic to fibroblast. Overall, the manuscript is suitable for publication after some necessary revisions.

1.     Figures should be revised extensively. Some of the figure was cut too much and some of the figure was duplicated. E.g. Fig.2a (top); Fig3 is duplicated.

2.     The authors were suggested to compare the FTIR spectrum of PITAU/Alg with pure PITAU and Alg.

3.     What is the formulation of PITAU/Alg in Table 1.

4.     The title of Abscissa was not in the meddle in Fig 7.

5.     As we know, the mechanical strength of DN hydrogel should be very high. Can the authors explain why the storage modulus of the DN hydrogel in this work was only about 300Pa?

6.     The format of Figure 10 is not in constant with other figures. The authors are suggested to redraw this figure using a constant format.

7.     Can “>75% cell viability” be considered biocompatible?

8.     More references should be cited in this paper. Lots of advanced hydrogels have been developed in recent years.

Author Response

(The authors gave the same response as above.)
